# Parvovirus (B19) Infection during Pregnancy: Possible Effect on the Course of Pregnancy and Rare Fetal Outcomes. A Case Report and Literature Review

**DOI:** 10.3390/medicina58050664

**Published:** 2022-05-15

**Authors:** Dovile Kielaite, Virginija Paliulyte

**Affiliations:** 1Faculty of Medicine, Vilnius University, 03101 Vilnius, Lithuania; 2Center of Obstetrics and Gynecology, Faculty of Medicine, Institute of Clinical Medicine, Vilnius University, 08661 Vilnius, Lithuania; Virginija.Paliulyte@santa.lt

**Keywords:** congenital infection, hydrocephalus, parvovirus B19, pregnancy, subdural hematoma

## Abstract

Infection caused by human parvovirus B19 (B19) often has mild yet wide-ranging clinical signs, with the course of disease usually defined as benign. Particularly prevalent in the population of young children, the virus is commonly transmitted to the parents, especially to susceptible mothers. During pregnancy, particularly the first and second trimesters, parvovirus infection can lead to pathology of the fetus: anemia, heart failure, hydrops, and disorders of physical and neurological development. In severe cases, the disease can result in fetal demise. This article presents a rare case of manifestation of B19 infection during pregnancy. At the 27th week of gestation, a sudden change in fetal movement occurred in a previously healthy pregnancy. The examination of both fetus and the mother revealed newly formed fetal subdural hematoma of unknown etiology and ventriculomegaly. Following extensive examination to ascertain the origin of fetal pathology, a maternal B19 infection was detected. Due to worsening fetal condition, a planned cesarean section was performed to terminate the pregnancy at 31 weeks of gestation. A preterm male newborn was delivered in a critical condition with congenital B19 infection, hydrocephalus, and severe progressive encephalopathy. The manifestation and the origin of the fetal condition remain partially unclear. The transplacental transmission of maternal B19 infection to the fetus occurs in approximately 30% of cases. The main method for diagnosing B19 infection is Polymerase Chain Reaction (PCR) performed on blood serum. In the absence of clinical manifestations, the early diagnosis of B19 infection is rarely achieved. As a result, the disease left untreated can progress inconspicuously and cause serious complications. Treatment strategies are limited and depend on the condition of the pregnant woman and the fetus. When applicable, intrauterine blood transfusion reduces the risk of fetal mortality. It is crucial to assess the predisposing factors of the infection and evaluate signs of early manifestation, as this may help prevent the progression and poor outcomes of the disease.

## 1. Introduction

Parvovirus B19 was first identified by Cossart et al. in 1974. The nonenveloped 20–25 nm single-stranded DNA virus belongs to the *Parvoviridae* family and is known to be the only representative pathogenic to humans. B19 is especially cytotoxic to erythrocyte precursor cells and cardiomyocytes [1,2,3]. The virus spreads through the respiratory pathway but can also be transmitted vertically from the mother to the fetus through bone marrow and blood transfusion or organ transplantation [4,5,6]. B19 infection is usually mild and benign; it manifests with a broad range of clinical symptoms: infectious erythema, arthropathy, acute aplastic crisis, cytopenia, and more. In severe cases during pregnancy, hydrops of the fetus may cause intrauterine death to occur [5,7]. The infection causes anemia, hypoalbuminemia, hepatitis, and myocarditis, which increase the risk of heart failure and non-immune hydrops fetalis. It is important to remark that as many as 15–20% of fetal non-immune hydrops cases are caused by B19 infection. With an active infectious process present during pregnancy, spontaneous recovery is possible, but there are known cases of fetal demise [5]. The matter of B19 infection during pregnancy is problematic due to unspecific clinical signs, late or missed diagnoses, and potentially grave complications of the disease. Thus, knowledge about the infection can help prevent the disease and avert fetal damage and the risk of possible life-long complications [8]. The primary prevention of infections during pregnancy is crucial in the task of preventing congenital infections in neonates. Vaccines are effective in many but not all mother-to-child transmitted infectious diseases. When vaccines are not an option, individual and public knowledge directly correlates to the successful prevention of congenital infections and long-term consequences [9]. Special attention ought to be addressed to the promotion of public health to raise awareness of community-acquired diseases such as B19 infection. It is common knowledge that prevention is better than intervention, especially considering pregnancy-related issues [10]. During the parvovirus epidemic, pregnant women are advised to be aware of foci of infection and to comply with personal hygiene requirements [11]. Medical professionals are recommended to actively monitor the dynamics of the condition of pregnant women with a more increased risk of infection. Education on B19 infection during pregnancy may help patients identify the first signs of the disease and encourage them to consult with healthcare professionals on time.

## 2. Case Report

A 31-year-old pregnant woman at the 27th week of a previously uncomplicated pregnancy experienced a sudden change in fetal movement. On the day of = admission, the patient’s condition was adequate, and fetal movements were absent. The tones of the fetus’s heart were rhythmic and clear, the heart rate normal (150 bpm), and the nonstress test was reactive. Gynecological examination implied no visible pathology or premature labor. Laboratory tests of the mother revealed mild-grade anemia (Hb 10.6 mg/dL), decreased hematocrit (0.285 L/L), and decreased erythrocyte count (3.08 × 10^12^/L). SARS-CoV-2 infection was not detected.

An ultrasound examination of the fetus was carried out through the abdominal wall, during which, pathological changes in the fetal brain were first observed (Figure 1, Figure 2 and Figure 3). A suspected tumor-like mass (2.75 × 2.16 cm) was seen in the anterior cerebral lobe of the right hemisphere, and dislocation of middle line and mild ventriculomegaly (1.38 cm) of the left lateral ventricle of the brain was present. Magnetic resonance imaging (MRI) of the fetal brain was carried out to establish the diagnosis (Figure 4).

The clinical manifestation of the disease and fetal sonogram findings were highly indicative of congenital fetal infection, and tests for congenital TORCH (Toxoplasmosis, other agents Rubella, Cytomegalovirus and Herpes Simplex) infections were carried out. All TORCH tests were negative.

The parvovirus infection was confirmed using molecular tests. Further outpatient care was recommended with the self-monitoring of fetal movements. Continuous outpatient visits to assess the dynamics of the fetal condition were advised.

Five days after the initial diagnosis, the patient complained of an aching sensation in the lower abdomen, but during the physical examination, no pathological signs were detected. Fetal movements were normal. During the physical evaluation, the height of the uterine fundus corresponded to the duration of pregnancy, and there were no signs of growth retardation or amniotic fluid abnormality. Left middle cerebral artery (MCA) pulsatile index (PI) = 53.8; right MCA PI = 39.19. The cerebral cortex on the left side was 0.7 cm, the subdural hematoma on the right measured 6.14 × 1.44 cm in size. Midline structures of the brain were shifted to the left by 7.8 mm. There were no pathological vascular formations, and all blood vessels in the Willis circle were assessed as functioning. The condition did not indicate surgery and the immediate termination of the pregnancy; hence, it was recommended to continue regular weekly ultrasound scans for ventriculomegaly and to conduct an outpatient MRI test after two weeks.

At 29 weeks and 2 days of pregnancy, an ultrasound of the fetal brain showed a slightly more enlarged left lateral ventricle and increased pressure on the cerebral cortex in the left hemisphere due to ventriculomegaly. The subdural hematoma did not change significantly, including no new bleedings. The measurement of the biparietal diameter of the head of the fetus was larger than normal (>90th percentile), as was the circumference of the head (>90th percentile).

One week later, the left lateral ventricle size increased to 2.7 cm. The MCA PI values on the left and right sides decreased (45.1 and 39.3, respectively). The compression of the cerebral cortex progressed (0.43 cm). Subdural hematoma did not increase in retrospect (5.4 × 1.6 cm). The midline remained shifted to the left (7.5 mm). The biophysical profile of the fetus was 10 (10) points. The estimated weight of the fetus was 1769 g ± 258 g (75–90th percentile), and the dimensions of the head remained above the 90th percentile. Biometric data matched the gestational age. The umbilical artery systolic/diastolic ratio (S/D) and resistance index (RI) were within the normal range (2.17 and 0.59, respectively). PI was in less than the 5th percentile (0.75), fetoplacental blood circulation was normal, nonstress test (NST) reactive, fetal pulse—145 bpm, and the heart tones were clear and rhythmic.

A fetal MRI test at 31 weeks demonstrated negative dynamics compared to 3 weeks prior (Figure 5). The right subdural space significantly expanded, the compression of the right hemisphere of the brain increased, and an image of emerging hygroma was observed. Hydrocephalus, likely occlusive, with dilatation of the left lateral ventricle and compression/atrophy of the cerebral parenchyma of left hemisphere, developed. A decreased volume of cerebellar parenchyma due to signs of emerging external paracerebellar hydrocephalus was observed. The midline remained shifted to the left for 6 (8–9) mm.

Due to negative MRI changes, the woman was hospitalized in the pregnancy pathology department.

A multidisciplinary team with the participation of pediatric neurosurgeons decided to terminate the pregnancy, according to the negative dynamics in the fetal MRI. The main goal of this decision was to attempt urgent surgical decompression of the newborn brain to allow the possibility of slowing down the developing hydrocephalus and cerebral atrophy. The maturation of the fetal lungs with corticosteroids was completed, and a planned cesarean section was performed at 31 weeks and 3 days of gestation.

A premature male newborn, 1850 g, and 44 cm in length, with 9/9 on the Apgar scale after one minute, was delivered. The umbilical cord pH was within normal range (7.41). The newborn was born in critical condition with visible hypotonia and hyporeflexia. He was transferred to the Neonatology department for further examination and treatment.

A congenital infection of parvovirus B19 was diagnosed in the newborn without detected growth in the blood culture medium. The results of the histological examination of the placenta did not show significant changes. No inflammatory infiltration in the fetal coatings and umbilical cord were found.

A multidisciplinary medical team assessed the findings in the newborn’s brain as secondary bleeding and ventriculomegaly due to encephalomalacia. In the absence of brain compression symptoms after birth, active treatment methods that could potentially improve the condition of the baby at the time were excluded, and cerebral atrophy progressed. Oxygen therapy was required, and feeding through a gastroenteric tube was used, as the newborn was unable to swallow by himself. Almost two months after being born, the baby was discharged from the Neonatology Department in a stable condition with comorbid conditions caused by congenital B19 infection: seizures, congenital hydrocephalus, multiple complications of preterm birth, and gastrostomy. Due to the incurable condition, symptomatic treatment was suggested. To manage neurological symptoms, 5 mg of Phenobarbital three times per day was prescribed. Vitamin D, Ferrous Bysglicinate Chelate, and Sodium Alginate supplements were given. Passive physical and massage therapy were administered. Oxygen therapy and nutrition through a gastrostomy tube were required. Palliative care was provided. As the pathology progressed, the aggravation and complications of the disease led to the death of the child at the age of 4.5 months.

## 3. Discussion

In the clinical case described, a sudden change in the movements of the fetus of unknown etiology was observed during the second trimester in a previously healthy pregnancy. Objective findings showed subdural hematoma, hydrocephalus, progressing fetal cerebral leukomalacia, ventriculomegaly ex vacuo, cerebral atrophy, and maternal subclinical manifestations of parvovirus B19 infection. Due to the severe condition of the fetus, a cesarean section was performed at 31 weeks of gestation. A male newborn was delivered with congenital B19 infection, congenital hydrocephalus, cerebral immaturity, cerebrovascular pathology, and conditions related to preterm birth.

Overall, cases of extremely severe fetal conditions and negative outcomes of B19 infection during pregnancy as reported in our patient are not widely recorded, explaining our interest in the case. Additionally, cases of intracranial hemorrhage of the fetus during pregnancy are rare in the scientific literature. Interestingly, the differentiation of the disease of the fetus and the origin of acute pathological changes were complicated by the patient’s unusual history: a rare tumor—a benign spinal cord hemangiopericytoma—was diagnosed and surgically removed three years before pregnancy. As it is suggested, fetal congenital brain tumors can potentially cause spontaneous intracranial hemorrhages in utero [12,13]. Considering previous maternal hemangiopericytoma and its tendency to reoccur, a possibility of a rare metastatic fetal neoplasm or a tumor-associated hemorrhage was plausible [14]. Nevertheless, ultrasound examination and MRI imaging did not unveil an underlying tumor. Extensive examination revealed primary signs of congenital fetal infection, and other possible causes of fetal intracranial hemorrhage (such as alloimmune thrombocytopenia, physical trauma during pregnancy, maternal anticoagulant use, vitamin K deficiency, as well as mutation of the COL4A1 gene) were excluded, and the etiology of fetal pathology was distinguished.

The analysis of the presented clinical case and the description of the manifestation are useful for a better understanding of the consequences of the B19 infection, the prediction of the course of the disease, and the importance of possible prevention. Parvovirus B19 usually spreads through respiratory secretions and clinically occurs after a one-to-three-week incubation period. Significantly, the infection can also be transmitted vertically from the mother to the fetus through bone marrow, organ transplantation, and the transfusion of blood products [4,5,6].

Although B19 infection is usually mild and benign, it can manifest in a wide range of clinical symptoms that may cease spontaneously or progress to serious conditions and death. The prodromal period is characterized by fever, nausea, headache and muscle pain, rhinorrhea, weakness. Clinically active disease is often manifested by erythema, arthropathy (in middle-aged women, joint damage occurs in 50% of cases), acute or chronic anemia, myocarditis, and lethal cytopenia. Infection during pregnancy can also cause intrauterine fetal death. Nevertheless, more than half of the infections occurring in pregnant women are asymptomatic [5,15,16,17]. This aspect made our case more difficult because the patient did not experience symptoms of infection until the change in fetal movements. As reported in the literature, the risk of acquiring the infection increases when having frequent contact with children. Women raising infected children are most often susceptible to infection, making up around 50% of reported cases [4,15]. More than half of the adult population is considered to have had B19 infection during their lifetime, although significant differences have been observed in individual age groups. According to the literature, specific antibodies (IgG) against the B19 virus were detected in up to 20% of children under 5 years of age, in up to 40% of minors aged 5–18 years, and in up to 80% of the adult population [18]. In Europe, seroconversion in the children and adolescent population is measured at up to 50–70% and stays similar in young adulthood years. Nevertheless, the prevalence is higher for individuals from 25–30 years of age [18,19]. It should be noted that most cases of active B19 infection are recorded in children aged 5–15 years; therefore, parents of minors or individuals working in educational institutions are considered to have an increased risk of acquiring the disease [5]. As our patient has a young child, her risk of contracting parvovirus infection was higher. According to the recent scientific literature, in Europe, around 40% of pregnant women are susceptible to B19 infection [15]. Parvovirus affects 1–5% of pregnant women, [3,12] and the risk of transmission to the fetus is 25–30% [16,17,20].

After acquiring the infection, pregnancy and childbirth may progress normally, but there is a 5–10% chance of severe fetal pathology. Severe congenital abnormalities occur rarely, since the virus is not considered to be a significant teratogen. B19 infection can infrequently cause damage to the fetal brain and disorders of neurological development, especially if the infection is transmitted within the first 20 weeks of gestation. The virus is regarded to be an important cause of fetal loss, especially in the first trimester, with reduced risk of fetal demise in the second half of the pregnancy [3,16]. The fetus is known to be more susceptible to parvoviral infection during the first and second trimester, and this is related to the development of erythroid precursors. The risk decreases in the third trimester of the pregnancy when the lifespan of erythrocytes elongates, hematopoiesis takes place in the bone marrow, and the fetus exhibits an immune response to the viral infection. Thus, the risk of poor fetal outcome is elevated during the first and second trimesters of the pregnancy but is existent during the third trimester nonetheless [21,22,23].

The risk of transmission to the fetus is most eminent during the first and second trimesters, and it occurs within 12 weeks after the mother gets infected. This is explained by a decrease in the concentration of placental antigens (P–Ag), as the gestational age of the fetus increases. Although the mechanism is not completely clear, P–Ag is presumed to be significant in the transfer of the virus through the placenta [20]. For our patient, the infection was detected in the 27th week of pregnancy, during the second trimester, while in search for the condition that caused the subdural hematoma in the fetus. In general, fetal subdural hematoma is a rare condition, especially in the absence of a traumatic injury in current pregnancy. Fetal coagulation disorders (e.g., alloimmune thrombocytopenia, a maternal-anticoagulant-use-related condition) and minor maternal injuries are common causes of intracranial bleeding in the fetus. Hemorrhages are less commonly caused by ischemic hypoxic damage and anemia, but in as many as 47% of cases, the origin of hemorrhages remains unknown [24,25]. In the case of our patient, the named disorders that could provoke the development of a subdural hematoma were not identified. For this reason, the fetal condition was associated with the pathophysiological mechanism of detected B19 infection, excluding other possible causes.

Once in the fetal system, the virus can affect multiple organs and cause severe anemia, hypoalbuminemia, hepatitis, myocarditis, heart failure, and non-immune hydrops. However, a negative effect on the neurological development of the fetus is rarely described in the literature. It is important to note that the consequences of B19 infection during pregnancy can be seen not only in the fetus (in addition to previously mentioned conditions, spontaneous abortion, intrauterine growth retardation, and fetal demise are possible) but in the infant as well [8]. Parvovirus infection is associated with anomalies in the brain, eyes, and cardiovascular and gastrointestinal systems, although these are generally rare [26]. Long-term neurological defects mainly due to B19-related fetal hydrops can occur in about 9.8% of the cases [23]. Other long-term complications of fetal B19 infection include hepatic insufficiency, anemia, and myocarditis [27]. In our case, the neonate presented with severe complications related to congenital B19 infection that are said to be uncommon in the scientific literature. Grave progressing neurological complications of B19 infection lack effective treatment methods; thus, palliative treatment and care were provided. Interestingly, B19 infection causes as many as 15–20% of cases of non-immune hydrops and is one of the most common etiological factors of the condition. Therefore, in the case of suspected hydrops fetalis, it may be relevant to carry out serological tests on the mother and/or fetus for possible parvovirus infection [28,29].

The possible consequences of the disease and clinical manifestations are explained by the pathophysiological mechanism of the viral effect. Parvovirus targets human erythrocyte precursor cells in the bone marrow and fetal liver in early pregnancy, while cardiomyocytes are affected by the cytotoxicity of expression of the viral unstructured protein [3,5,16,30]. Since active hemopoiesis occurs during the first and second trimesters of pregnancy, large amounts of erythrocyte precursors mature, and the fundamental body systems develop, B19 infection is associated with worse fetal outcomes when acquired in the first half of the pregnancy [16,20,28]. Acute anemia develops due to hematopoietic cell apoptosis caused by exposure to the virus and the disruption of erythropoiesis. Anemia might resolve spontaneously or progress to cause heart failure, hydrops, and, seldom, intrauterine fetal death. Nevertheless, spontaneous recovery or subclinical manifestation that does not affect the condition of the woman and the fetus or the outcome of pregnancy is possible [3,5,31]. Interestingly, our patient was diagnosed with pregnancy-related anemia at 16 weeks of gestation, and approximately 11 weeks later, a subdural hematoma was found in the fetus. As discussed earlier, the infectious process is characterized by anemia due to the high susceptibility of viral molecules to erythrocyte precursor cells. It is possible that anemia, thought to be a pregnancy-related condition, may have concealed a manifesting B19 infection, after which, in 12 weeks, the transmission of the infection to the fetus theoretically occurs. Our patient experienced a chronological sequence of clinical manifestations that coincide with the one described in the literature.

Infection is generally presumed based on clinical findings and confirmed by serological tests: IgG seroconversion and/or IgM in the blood sample and serum PCR test. In our case, the patient tested positive in the parvovirus PCR test, indicating circulating viral DNA. Viral nucleic acid detected in blood serum can suggest transient infection, active viral replication, or residual viral DNA. Additional serological IgM tests could have provided more information about the state of the disease, since IgM is typically found within one to three weeks of infection and persist for about two to three months [5,7,32].

Diagnostics can be challenging because, in the fetus, viral particles can only be detected at the stage of viremia [33]. The test for possible fetal B19 infection is a PCR test on a specimen collected during cordocentesis or amniocentesis with amniotic fluid, with PCR tests having sensitivity higher than 97% and specificity of 79–99% [7,34]. B19 DNA may be detected in a blood sample of a newborn with congenital infection, but it is important to assess the likelihood of a false negative result [35]. Some sources suggest that the most accurate method for detecting congenital infection after birth is a PCR test performed on the newborn’s bone marrow [34]. In our case, a PCR test was performed on the newborn’s blood serum. The newborn tested positive for a parvovirus PCR test; therefore, congenital infection was confirmed.

The differential diagnosis of B19 infection is particularly important to better assess the severity of the patient’s condition, the possible course of the disease, treatment, and outcomes. For pathological changes associated with neurological complications (in our case—intracranial hematoma), an antenatal ultrasound examination is extremely accurate and relevant. Meanwhile, a Dopplerometric test for the maximum systolic velocity of the blood flow of the middle cerebral artery (MCA) is considered exceptionally sensitive to diagnose fetal anemia [25,33]. After seroconversion in pregnant women within the first 20 gestational weeks, non-invasive ultrasound monitoring of fetal anemia is routinely provided. If fetal anemia or hydrops of unknown origin is previously diagnosed, amniocentesis is indicated for the detection of suspected fetal infection [36,37]. Ordinarily, routine screening for infection in low-risk pregnancies is not recommended. An urgent examination for possible infection is advised after confirmed placentomegaly or hydrops during an ultrasound or after close contact with an infected individual. In our case, placentomegaly was not found, as the placenta’s size was according to gestational age (366 g, 15 × 14 × 2.8 cm at birth). The histological evaluation of the placenta, umbilical cord, and fetal membranes showed no signs of inflammation or pathology. It is useful to perform cordocentesis to determine the values of fetal hemoglobin and reticulocyte count, and, after evaluating the findings, to prescribe the most appropriate treatment method [3,38]. 

The treatment of infection depends on the condition of the pregnant woman and the fetus. As the time of childbirth or the premature termination of the pregnancy approaches, corticosteroids are recommended for the maturation of the fetal lungs [3,39]. Intrauterine transfusion, considered a standard treatment method that significantly reduces the risk of fetal mortality, is mostly used to treat infection-specific, severe fetal anemia [3,36].

Our patient received iron supplements during pregnancy due to mild-grade anemia. Because there were no indicators of fetal anemia and no effective treatment methods present, monitoring of the condition was decided as the best course of treatment. Care tactics were chosen by the existing methodologies and recommendations of obstetrics and gynecology. As the condition of the fetus worsened, it was decided to perform a planned cesarean section, before which, our patient was prescribed corticosteroids for fetal lung maturation. To reduce brain compression, developing hydrocephalus, and cerebral atrophy, the removal of the subdural hematoma with the decompression of the fetal brain was indicated.

Based on statistical data and the latest scientific literature, it can be said that B19 infection during pregnancy can become an etiologic factor in severe fetal pathology. With an active infectious process during pregnancy, spontaneous recovery is possible, but there are known cases of fetal demise.

Information regarding the risk factors for infection (Table 1) and prevention, aspects of pathogenesis, and clinical manifestations can help prevent the progression of the disease and the consequences of late diagnosis. 

## 4. Conclusions

Parvovirus B19 infection during pregnancy rarely causes severe impairment of the health of the mother or her fetus, but there are known cases of lethal pathology to the unborn baby or neonate. The complications of the inconspicuously progressive disease can be acute and difficult to predict and lack effective treatment methods. The virus can affect multiple organs and cause severe anemia, hypoalbuminemia, hepatitis, myocarditis, heart failure, and non-immune hydrops. Severe neurological complications of the infection are infrequent but possible, with fetal subdural hematoma being a rare and dangerous manifestation that, in our case, caused the child’s death after being born. It is important to consider the elevated risk of the B19 infection when both typical clinical symptoms and significant risk factors are present. In the absence of specific treatment and prophylactic measures, it is significant to constantly assess the progression of pregnancy in susceptible women with high risk factors. Considering the possibility of serious adverse effects of B19 infection, prevention is of importance. It is necessary to encourage pregnant women to be cautious during B19 outbreaks and comply with personal hygiene requirements.

## Figures and Tables

**Figure 1 medicina-58-00664-f001:**
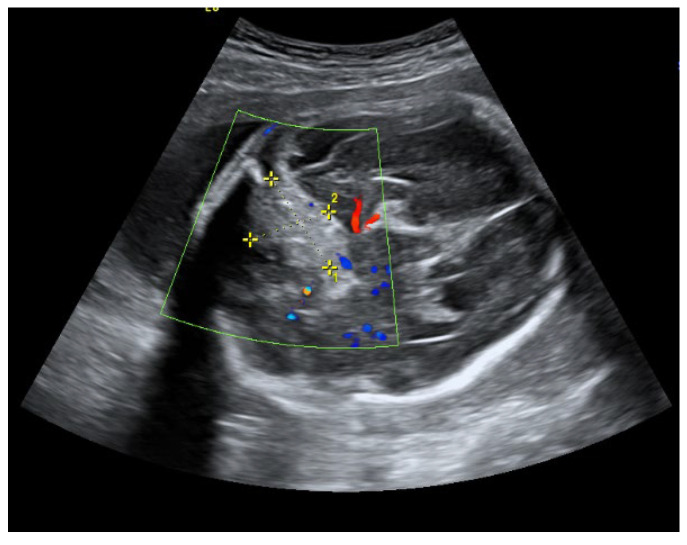
A suspected tumor in the right anterior fossa of the fetal brain.

**Figure 2 medicina-58-00664-f002:**
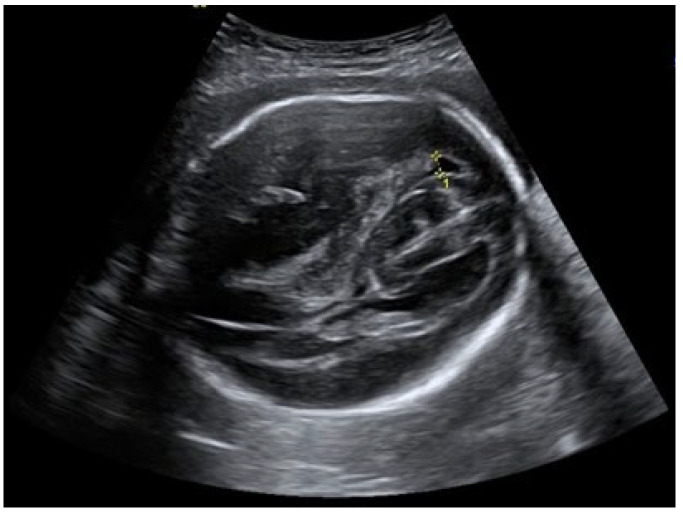
Right lateral ventricle and dislocated middle line of fetal brain.

**Figure 3 medicina-58-00664-f003:**
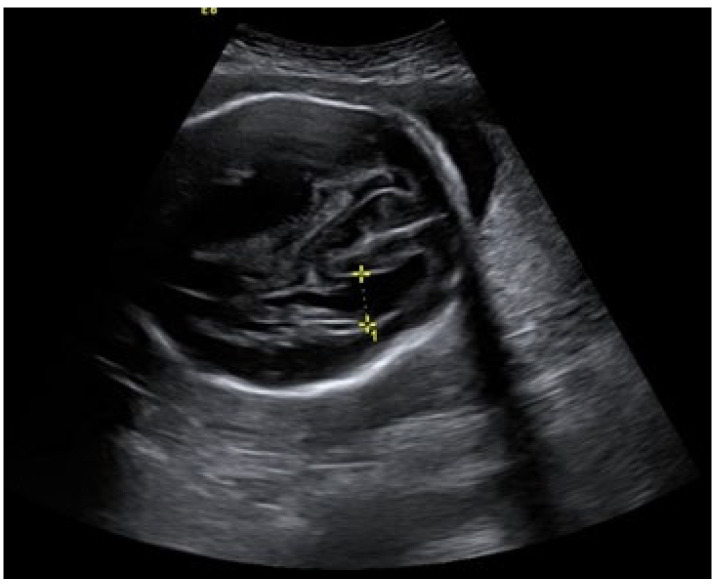
Ventriculomegaly of the left lateral ventricle.

**Figure 4 medicina-58-00664-f004:**
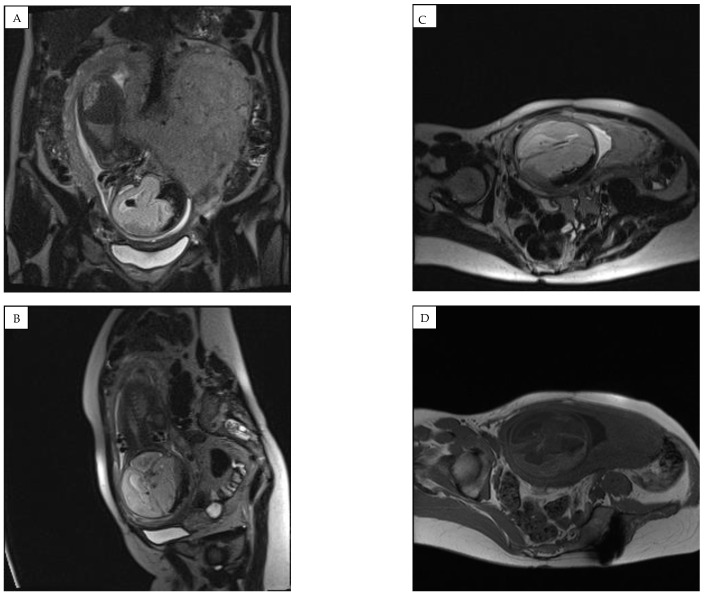
Magnetic resonance imaging (MRI) of the fetal brain was conducted to confirm the diagnosis of expansion of the right subdural space. (**A**) coronal plane and (**B**–**D**) saggital plane magnetic resonance images showing a hemorrhage with compression of the right hemisphere and a shift to the left of the midline structures was found, along with an accumulation of hemorrhagic signals in the right parasellar part.

**Figure 5 medicina-58-00664-f005:**
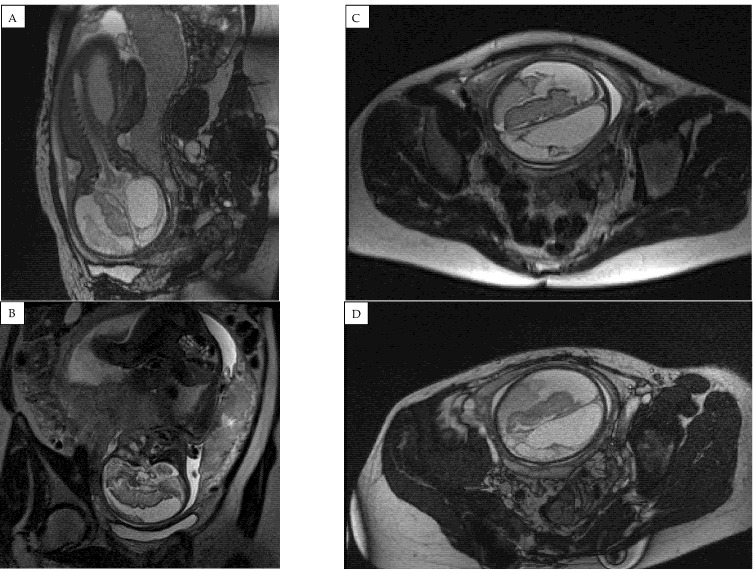
Fetal magnetic resonance imaging (MRI) findings at 31 weeks of pregnancy showing significantly enlarged right subdural space and hygroma formation. (**A**) saggital, (**B**) coronal and (**C**), (**D**) transverse plane magnetic resonance images showing an occlusive hydrocephalus with dilatation of left ventricle and compression of cerebral parenchyma of left hemisphere with decreased cerebellar parenchyma.

**Table 1 medicina-58-00664-t001:** Parvovirus B19 risk factors and symptoms.

Factors Increasing the Risk of B19 Infection
Previous or potential contact with B19 infection
Young children in the family (especially 5–7 years old)
Working with young children or in an educational institution
Working in a laboratory
Active B19 epidemic in living environment (early spring and winter seasons)
Siblings in the family
Individual aged up to 20 years and over 35 years old
Hematological disorders (anemia, etc.)
Disorders of the immune system
**Clinical findings and symptoms indicating a potential B19 infection**
Findings in an ultrasound examination of the fetus: 1. Placentomegaly 2. Hydrops fetalis 3. Fetal hydrocephalus 4. Calcifications in fetal brain 5. Fetal anemia
Erythema
Arthropathy (arthralgia or arthritis)
Thrombocytopenia
Acute anemia
Newly manifested: 1. Hepatitis 2. Myocarditis 3. Vasculitis Meningoencephalitis

## Data Availability

Not applicable.

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
