# Peer review of "Parvovirus (B19) Infection during Pregnancy: Possible Effect on the Course of Pregnancy and Rare Fetal Outcomes. A Case Report and Literature Review"

_medicina, 2022, doi:10.3390/medicina58050664_

Round 1

Reviewer 1 Report

Abstract:

(Transplacental transmission of maternal B19 infection to the fetus occurs in approximately 30% of cases. The main method for diagnosing B19 infection is Polymerase Chain Reaction (PCR) performed on a blood serum. In the absence of clinical manifestations, early diagnosis of B19 infection is rarely reached. As a result, the disease left untreated can progress inconspicuously and cause serious complications. Treatment strategies are limited and depend on the condition of the pregnant woman and the fetus. When applicable, intrauterine blood transfusion reduces the risk of fetal mortality. It is crucial to assess the predisposing factors of the infection and evaluate signs of early manifestation as it may help prevent the progression and poor outcomes of the disease.)

               This part not to be included in abstract. These things to be included in Introduction/ or in discussion part. Significance of the study or any innovative treatment or specific findings you want to highlight need to be written in abstract.

Introduction

It is important to remark that as much as 15–20% of fetal non–immune hydrops cases are caused by B19 infection. With active infectious process present during pregnancy, spontaneous recovery is possible, but there are known cases of fetal demise [5].

So, if so many cases are there with B19 infection during pregnancy, why are you calling it a rare presentation?

Case report

In a previously healthy pregnancy. Sentence needs modification

mild–grade anaemia (Hb 106 g/l)- needs to be corrected as 10.6gm/dl

Sars–CoV–2;  SARS–CoV–2

The maturation of the fetal lungs was completed: how you know that?

 44 cm, with 9/9 on the Apgar scale: HC 44cms? 9/9 Apgar score is of 1min or 5minutes?

The newborn was born in critical condition with visible hypotonia and hyporeflexia. When APGAR 9/9, how the newborn will be critical.

Gastrostomy: Why this procedure done to the baby? Describe.

As the pathology progressed, the aggravation and complications of the disease led to the death of the child at the age of 4.5 months.

What is the cause of death? Course of treatment provided in this 4.5 months to the baby need to be elaborated.

Discussion

Additionally, cases of intracranial haemorrhage of the fetus during pregnancy are rare in scientific literature.

Have you excluded the causes of intracranial haemorrhage of the fetus https://www.ncbi.nlm.nih.gov/pmc/articles/PMC5685119/

https://www.ncbi.nlm.nih.gov/pmc/articles/PMC5685119/table/t1/?report=objectonly

Parvovirus affects 1–5% of pregnant women [3,12] and the risk of transmission to the fetus is 25–30% [12,13,16].

Narrate the outcome Parvovirus affects in infants if mother have B19 during pregnancy. So that you could express your rarity if any.

 if the infection is transmitted within the first 20 weeks of gestation: describe the impact of B19 in fetus/neonate as per trimester infection to mother.

Fetal coagulation disorders (e.g., alloimmune thrombocytopenia, maternal anticoagulant use related condition) Have you excluded the factor deficiencies in the newborn?

 It is possible that anaemia, thought to be a pregnancy–related condition, may have concealed a manifesting B19 infection, after which, in 12 weeks the transmission of the infection to the fetus theoretically occurs. In your case, when the mother got infected with B19?

In our case, the patient tested positive for the parvovirus PCR test, indicating circulating viral DNA.

 What about IgG & IgM level? When the fetus got the infection from the mother? Acute / chronic?

Conclusions

 Parvovirus B19 infection during pregnancy rarely causes severe impairment of the health of the mother or her fetus, but there are known cases of lethal pathology to the unborn baby or neonate- from one case how can you say this?

Nothing mentioned about fetal outcome of B19 in conclusion, which you mentioned in the title.

Reviewer 2 Report

I've appreciated your job, is a nice paper a nice case well written. Infections during pregnancy are an interesting topic, I've loved pictures and the whole case.

I just want to suggest to improve the introduction with the concept of the importance of women awareness regarding risks and way to prevent infections. Women awareness is the strongest way to reduce the rate of infections during pregnancy therefore I would suggest to introduce and cite the subsequent papers, related to the importance of women awareness in the improvement of compliance with recommendations regarding the most common infection during pregnancy and the most common obstetric complications awareness is the future of our job.

PMID: 28831237

PMID: 34627198 

Round 2

Reviewer 1 Report

Well revised. congratulations